# Effects of Low-Dose Atorvastatin on the Peripheral Blood Mononuclear Cell Secretion of Angiogenic Factors in Type 2 Diabetes

**DOI:** 10.3390/biom11121885

**Published:** 2021-12-15

**Authors:** Anna Wesołowska, Hanna Winiarska, Jakub Owoc, Magdalena Borowska, Joanna Domagała, Przemysław Łukasz Mikołajczak, Saule Iskakova, Grzegorz Dworacki, Marzena Dworacka

**Affiliations:** 1Department of Pharmacology, Poznan University of Medical Sciences, 60-805 Poznan, Poland; hwiniar@ump.edu.pl (H.W.); mborowska@ump.edu.pl (M.B.); joanna_domagala@o2.pl (J.D.); przemmik@ump.edu.pl (P.Ł.M.); mdworac@ump.edu.pl (M.D.); 2Department of Clinical Immunology, Poznan University of Medical Sciences, 60-805 Poznan, Poland; jowoc@ump.edu.pl (J.O.); gdwrck@ump.edu.pl (G.D.); 3Department of Pharmacology, Asfendiyarov Kazakh National Medical University, Tole Bi Str. 94, Almaty 050000, Kazakhstan; saule1@inbox.ru

**Keywords:** angiogenic factors, diabetes, hyperglycemia, peripheral blood mononuclear cells, statins

## Abstract

The aim of this study was to investigate the influence of statins on the secretion of angiogenesis mediators by the peripheral blood mononuclear cells (PBMCs) derived from patients suffering from type 2 diabetes. The study group comprised 30 participants and included: 10 statin-treated patients with diabetes, 10 statin-free diabetic subjects, and 10 statin-free non-diabetic individuals. PBMCs isolated from the blood were cultured in vitro in standard conditions and in an environment mimicking hyperglycemia. Culture supernatants were evaluated for VEGF, MCP-1, Il-10, and Il-12 by flow cytometry using commercial BD^TM^. Cytometric Bead Array tests. The secretion of VEGF, MCP-1 and Il-12 by PBMCs, cultured both in standard and hyperglycemic conditions, was significantly lower in the statin-treated patients with type 2 diabetes in comparison with the statin-free diabetic patients. Conversely, the secretion of Il-10 was higher in the statin-treated than in the statin-free diabetic patients. VEGF, MCP-1 and Il-12 levels in PBMCs supernatants from the glucose-containing medium were higher than those from the standard medium in each of the diabetic groups. The results of the study suggest that statins in low doses exhibit an antiangiogenic activity, reducing the secretion of potent proangiogenic factors, such as VEGF and MCP-1, and increasing the secretion of antiangiogenic Il-10 by PBMCs, also under hyperglycemic conditions characteristic for type 2 diabetes.

## 1. Introduction

Atherosclerosis-based complications constitute the major cause of increased mortality and morbidity among patients with type 2 diabetes. It is supposed that the progression of atherosclerosis is partly connected with the dysregulation of angio- and arteriogenesis [1]. It is generally agreed that plaque neovascularization renders lesions vulnerable, leading to eventual rupture [2], whereas compensatory or collateral arteriogenesis is a beneficial and physiological process occurring in response to occlusion ischemia [3].

Statins, i.e., drugs used commonly in clinical practice with well-estimated lipid-lowering activity, exert numerous pleiotropic effects which are not lipid dependent [4]. Most of these activities are particularly beneficial for patients suffering from type 2 diabetes mellitus [5]. It has been showed that statins stimulate angiogenesis, which is regarded as a potential approach in the treatment of coronary artery disease [6]. In contrast, another study has demonstrated that atorvastatin impairs the myocardial angiogenic response to chronic ischemia [7], or even interrupts angiogenesis [8]. Furthermore, it has been suggested that statins may affect the process of angiogenesis biphasic, depending on the administered doses [9,10]. However, the reasons for these ambiguous effects have not been fully clarified yet.

One of the hypothesized mechanisms of statins mediated angiogenesis is the impact on the peripheral blood mononuclear cell (PBMC) secretion of factors involved in this process, such as: vascular endothelial growth factor (VEGF), monocyte chemotactic protein 1 (MCP-1), interleukin 10 (Il-10), and interleukin 12 (Il-12). VEGF is a direct pro-angiogenic factor, acting directly on the vascular endothelial cells [11]. In fact, the role of VEGF in early stages of neovascularization has been associated with such processes as the increase in vascular permeability, stimulation of endothelial cell proliferation and migration, as well as the activation of proteolytic enzymes. Moreover, this proangiogenic factor is also vital in the later stages of angiogenesis, influencing maturation and stabilization of the newly formed vessels [12]. MCP-1 indirectly enhances neovascularization by mobilizing and transdifferentiating bone marrow monocyte lineage cells into endothelial cell-like cells [13]. Moreover, this molecule can also directly induce angiogenesis in the mechanism, which is not related to its monocyte/macrophage recruitment. Recently, it has been demonstrated that MCP-1 influences angiogenesis via a novel zinc-finger protein, i.e., MCP-1-induced protein (MCPIP) [14].

VEGF and MCP-1 are potent proangiogenic molecules, whereas Il-10 and Il-12 play a critical role in the inhibition of this process. Il-10 control of new vessel formation is connected mostly with the inhibition of VEGF and basic fibroblast growth factor (bFGF) synthesis through the macrophage inactivation [15]. On the other hand, the antiangiogenic activity of Il-12 is associated not only with the inhibition of the proangiogenic factor expression [16], but also with the stimulation of other antiangiogenic factors secretion, such as interferon-γ (IFN-γ) and interferon-γ-inducible protein (IP-10) [17]. Additionally, Il-12 is also recognized as the mediator of antiangiogenic immunomodulatory action of angiostatin [18]. What is more, both of these interleukins can inhibit the synthesis of matrix metalloproteinases, as well as restrain the activation of vascular muscle cells [19,20].

Although previous studies [6,21] suggested the link between statins and angiogenesis, the relationship between diabetes-induced metabolic conditions and statin-dependent angiogenesis has not been fully determined yet. Therefore, the aim of this study was to investigate the influence of statins on the secretion of angiogenesis mediators by the peripheral blood mononuclear cells derived from patients with type 2 diabetes, particularly in the presence of acute hyperglycemia.

## 2. Materials and Methods

### 2.1. Patients

The protocol of the study was approved by the Bioethics Committee of Poznan University of Medical Sciences (870/09) and a written informed consent was obtained from every participant. The study involved 20 patients with type 2 diabetes, including 10 subjects treated with statin atorvastatin 10–20 mg/day for at least 3 months prior to the study, 10 statin-free patients, and 10 statin-free non-diabetic individuals. Patients with type 2 diabetes were treated with metformin. All subjects were Caucasian. The participants were patients of general practitioner clinics in Poznan (Poland). A complete physical examination and a laboratory evaluation of every subject was performed. Type 2 diabetes was diagnosed according to EASD (European Association for the Study of Diabetes) criteria [22]. Patients with a history of infections, severe inflammatory diseases, renal failure, liver failure, heart failure, thyroid dysfunction, and anemia were excluded from this study.

### 2.2. Protocol and Measurements

The experiment was performed in duplicates. PBMCs from 30 participants were isolated from the heparinized blood by centrifugation over Ficoll-Hypaque gradients (Sigma, St. Louis, MO, USA). After washing, the cells were resuspended in RPMI 1640 medium (pH 7.2; Sigma) containing 10% heat-inactivated fetal calf serum (MasciaBrunelli, Milan, Italy), 2 mM glutamine (Sigma), and 200 mg/mL gentamycin (BioWhittaker, Walkersville, MD, USA). PBMCs were seeded at a concentration of 2 × 10^6^ cell/mL, 100 mL/well, in 96-well flat-bottomed plates and incubated with or without glucose (22.2 mmol/L). Phytohemagglutinin (PHA) was used as a mitogenic factor (in a final concentration of 2.5 μg/mL). Following 72 h of incubation at 37 °C in a 5% CO_2_ humidified atmosphere, supernatants were collected, aliquoted, and stored at −80 °C until analysis. PBMCs culture supernatants were evaluated for VEGF, MCP-1, Il-10, and Il-12 by flow cytometry method, using commercial BD^TM^ Cytometric Bead Array tests (Becton Dickinson, Franklin Lakes, NJ, USA), according to the manufacturer’s instructions. The complexes of bead-investigated factors and antibodies labeled with PE were acquired with a FACS Canto flow cytometer and further analyzed with FCAP Array^TM^ Software (Becton Dickinson, Franklin Lakes, NJ, USA).

1,5-anhydro-D-glucitol (1,5-AG)—a marker of postprandial hyperglycemia—was measured using a modified column enzymatic method [23]. HbA_1_c (normal range: 4.1%–6.0%) was assayed by immunoturbidimetric method (COBAS Integra 400/700/800) standardized according to IFCC [24]. Lipid profile was obtained using standard enzymatic methods (Total cholesterol, HDL, LDL—cholesterol esterase method; triglycerides—glycerophosphate oxidase method).

### 2.3. Statistical Analysis

Statistical analyses were performed using Statistica version 10.0 (StatSoft, Inc., Dell, Round Rock, TX, USA). Data are reported as a median, mean and standard deviation. The ANOVA test was used to analyze the differences between the groups. A *p* value ≤ 0.05 was considered to be significant.

## 3. Results

Comparing the lipid profile between the examined groups (Table 1), it was observed that the values of individual parameters are slightly better in patients receiving statins, compared with the diabetic patients not receiving these medications. The level of chronic hyperglycemia measured by HbA1c was comparable for both groups with diabetes—statin-treated and non-statin-treated. Similarly, the intensity of glycemic variability expressed by 1,5-AG values did not differ significantly between these groups, and indicated for the presence of acute hyperglycemic episodes.

Comparing PBMCs cultures at basic conditions (standard medium) (DM+S and DM and N-DM), it was demonstrated that VEGF concentration in cell supernatants was significantly lower in the statin-treated patients with type 2 diabetes (DM+S) compared with the statin-free diabetic patients (DM). Furthermore, in both diabetic groups (statin-treated and statin-free), VEGF secretions were significantly higher than in the non-diabetic patients (N-DM) (Figure 1). Similarly, significantly important relationships were also observed for MCP-1 (Figure 2) and Il-12 (Figure 3) secretion by PBMCs cultured in the standard medium. The secretion of Il-10 in the same basic conditions was significantly higher in the statin-treated than in the statin-free diabetic patients, and in these groups Il-10 concentrations were lower than in patients without metabolic deteriorations (Figure 4.).

Comparing PBMCs secretion in the presence of glucose (DM+S+Glu and DM+Glu and N-DM+Glu), the analogical differences were observed for each of the angiogenic factors: VEGF (Figure 1), MCP-1 (Figure 2), Il-12 (Figure 3), and Il-10 (Figure 4), as previously described for the standard medium.

Analyzing exclusively the influence of glucose on the secretion of angiogenic factors, it was found that VEGF (Figure 1), MCP-1 (Figure 2) and Il-12 (Figure 3) levels in PBMCs supernatants from the glucose-containing medium were higher than those from the standard medium in both diabetic groups: the statin-treated and statin-free patients. Additionally, the concentration of Il-12 in supernatants of PBMCs cultures pre-incubated with glucose from the non-diabetic subjects was also significantly higher when compared with the results obtained in the standard conditions (Figure 3). No differences were observed in Il-10 levels in supernatants of PBMCs cultures from each of the analyzed groups in relation to the presence of glucose in the culture medium (Figure 4).

## 4. Discussion

Plaque angiogenesis is an important factor contributing to the development of unstable lesions in the course of atherosclerosis [25]. Nevertheless, the ultimate effect of statins on angiogenesis has not been elucidated yet. The results of the previous studies are divergent. Both proangiogenic [26] and antiangiogenic [27] activity of statins has been observed. It was proposed that the opposing action of statins on the process of angiogenesis might be dose dependent. It was revealed that low doses of statins exhibit proangiogenic effects, whereas the use of high doses was associated with antiangiogenic activity [9,10]. Furthermore, the influence of statins on the progress of angiogenesis appears to be particularly important in patients with diabetes who are commonly treated with these drugs. It was demonstrated that angiogenesis is modified by diabetes-related mechanisms, and the progress of angiogenesis in the diabetic patients is slightly different compared with those without metabolic disorders [28,29]. It is generally accepted that angiogenesis occurring within vascular plaques in the diabetic patients may be more intense and dangerous than in individuals not suffering from diabetes. Moreover, its severity is strongly dependent on glycemic control [30]. Nevertheless, the exact step in the course of angiogenesis, which is affected by both statin-treatment and diabetes-related hyperglycemia, has not been exactly defined.

Peripheral blood mononuclear cells, including monocytes, may be one of the targets of statins, and the effects of statins on PBMCs may constitute the potential mechanism underlying early intervention for atherosclerosis. In fact, PBMCs play a crucial role in neovascularization, secreting numerous factors which modulate this process. PBMCs have been recognized as a source of many proangiogenic molecules, such as VEGF, MCP-1 [31], TNF-α (Tumor Necrosis Factor-α), Il-1, Il-6, Il-8, PDGF (Platelet-Derived Growth Factor), FGF (Fibroblast Growth Factor), TGF (Transforming Growth Factor) and antiangiogenic cytokines, such as Il-10, Il-12, and interferon [32]. Therefore, an imbalance between angiogenic inducers and inhibitors seems to be a vital factor in the pathogenesis of macro- and microvascular complications of diabetes mellitus [33]. Furthermore, the role of PBMCs in the angiogenic process seems to be important not only because of their ability to secrete angiogenic factors, but also due to their specific site of action—localization within vascular plaques. This, in turn, may suggest that these cells may significantly modulate the formation of new blood vessels within the atherosclerotic plaque located in the peripheral vessels, thus affecting the risk of vascular complications.

This study constitutes an attempt to clarify whether statins affect angiogenesis by regulating the secretion of pro- and antiangiogenic modulators by PBMCs from type 2 diabetic, as well as non-diabetic, patients, particularly in the presence of hyperglycemia, which is an important factor increasing the secretion of many proangiogenic factors [34]. Moreover, it has recently been revealed that especially acute postprandial hyperglycemic episodes are the determinants of proangiogenic factor levels in the serum of type 2 diabetic patients [35,36].

A number of clinical studies [28,37,38] indicated that plasma concentrations of proangiogenic factors were higher in the diabetic patients, particularly if accompanied by diabetic complications, compared with non-diabetic individuals. These findings were also supported by the results of small number in vitro studies [39] in which the secretion of proangiogenic factors by PBMCs was investigated. The results of our present research confirmed these earlier in vivo and in vitro observations, namely that diabetes increased the readiness of the peripheral blood cells to release proangiogenic factors (VEGF, MCP-1).

The next step in our research was to verify in which direction, and whether statins may affect angiogenic factor secretion by PBMCs at all. It was demonstrated in vivo that statins both in high [21] and low doses [35] reduce the serum concentration of VEGF in diabetic patients. On the basis of our in vitro studies, we can speculate that this statin-related decrease in VEGF plasma level is partially related to the inhibition of VEGF secretion by PBMCs. Moreover, this protective effect was observed in the cultured cells both under normal conditions, as well as in the presence of glucose in the medium. In accordance with our results, Ho et al. [40] also showed that the use of statins in the cultured mesangial cells decreased VEGF expression, which had previously been enhanced by glucose.

To date, there have been no studies concerning the influence of glucose on the VEGF secretion by PBMCs. Khatri et al. [41] demonstrated that ROS (Reactive Oxygen Species) promoted VEGF expression in smooth muscle cells (SMCs). Although this study did not refer to PBMCs, it clearly indicated that VEGF production was strongly associated with free radicals, undoubtedly connected with the presence of hyperglycemia. In fact, this concept can also partly account for the mechanism by which statins, referred to as antioxidative agents [4], reduce glucose-dependent VEGF secretion.

It was revealed earlier that simvastatin treatment significantly reduced the monocyte chemoattractant protein 1 (MCP-1)-induced monocyte migration [42]. The results published by our group [36] revealed that the concentration of MCP-1 in the diabetic patients was reduced by statins. In contrast, Rabkin et al. [43] did not confirm the significant influence of statins on MCP-1 concentration in the serum of the diabetic patients. Our present in vitro study confirmed the results derived from in vivo studies, showing that glucose stimulated MCP-1 secretion by PBMCs, and that statins were able to decrease the intensity of that process. The ability of statins to inhibit MCP-1 synthesis by PBMCs was confirmed by Romano et al. [44], but contrary to our experiment, that study was devoid of the glucose stimulation phase. Glucose, and, in particular, short-lasting hyperglycemia episodes, is potentially involved in the increased MCP-1 synthesis by PBMCs by means of NF-κB activation [45], as well as an increased generation of free radicals [46]. The statin-related effect, the inhibition of HMG-CoA reductase, which is involved in the NF-κB signaling pathway [47], may be responsible for the suppression of MCP-1 secretion induced by these drugs. This hypothesis was supported by Diomede et al. [48] in an in vivo experiment, which demonstrated that the inhibition of mevalonate synthesis may influence the MCP-1 production by hepatocytes.

Il-10, secreted among others by macrophages, is recognized as a cytokine presenting antiangiogenic [15], anti-inflammatory, and antiatherogenic, as well as diabetes-protective, action [49]. Our results, which revealed that Il-10 secretion by PBMCs is lower in diabetic patients with high risk of atherosclerotic complications compared with the healthy controls, seem to be reliable. However, the possible influence of a statin treatment on the secretion of Il-10 remains unclear. It was previously suggested that statins were able to increase plasma level of Il-10 [50]. Meng et al. [51], in an animal (mice) experiment, observed that the drugs from this group significantly increased Il-10 concentration in atherosclerotic plaques, although that study did not concern type 2 diabetes. Our in vitro research results showed that HMG-CoA inhibitors increased the secretion of that antiangiogenic factor by PBMCs derived from the diabetic patients. Contrary to our results, Ferrari et al. [52] observed that treatment with statins decreased the secretion of Il-10 by PBMCs. On the other hand, Naoumova et al. [53] observed that treatment with statins did not alter Il-10 mRNA in the mononuclear cells in patients with hypercholesterolemia.

Surprisingly, the results of our present study demonstrated that secretion of antiangiogenic Il-12 by PBMCs was higher in patients with type 2 diabetes compared with the healthy subjects. However, taking into consideration not only the antiangiogenic, but primarily the pro-inflammatory effects of Il-12 [54], our results seem to be reliable. Moreover, our study shows that pre-incubation with glucose was associated with an increase in Il-12 secretion by PBMCs. These results are consistent with the previous study indicating glucose-dependent activation of protein kinase C, p38 MAPK (p38), c-Jun terminal kinase, and inhibitory-κB kinase, resulting in the Il-12 secretion by macrophages [55]. It is worth noting that in the statin-treated patients with type 2 diabetes, the concentration of Il-12 in supernatants was significantly lower compared with the statin-free diabetic patients, independent of the medium type. These results are consistent with the previous observations that statins inhibit the production of certain cytokines (including Il-12) by means of inhibiting the secretion of isoprenoids [4].

The results of our in vitro study suggest that statins in low doses exhibit rather antiangiogenic than proangiogenic activity. Certainly, statins reduce the secretion of potent proangiogenic factors, both direct (VEGF) and indirect (MCP-1), and increase the secretion of the typical antiangiogenic factor (Il-10). It seems particularly vital to bear in mind that these statin-related protective effects were observed even under acute hyperglycemic conditions. The reducing impact of statins on the secretion of the antiangiogenic Il-12 raises certain doubts; nevertheless, considering its pro-inflammatory action, the results appear to be justified.

The potential antiangiogenic activity of statins in diabetes, via affecting the secretion of angiogenic factors by PBMCs as suggested by our study results, can be extremely beneficial. It may lead to a reduction in angiogenesis within the blood vessel walls, preventing the atherosclerotic plaque rupture, and finally leading to a reduction in serious complications of diabetes.

The limitation of the study is the relatively small number of subjects in each group; however, standard conditions provided for all procedures of the experiment should be considered as factors compensating these limitations.

## Figures and Tables

**Figure 1 biomolecules-11-01885-f001:**
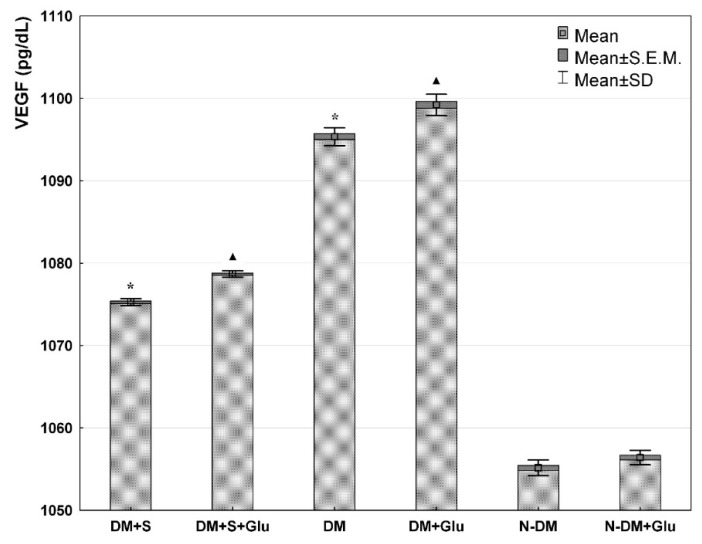
VEGF level in the supernatant from a culture of human PBMCs isolated from statin-treated or statin-free type 2 diabetic patients, or from non-diabetic controls in the medium only, or preincubated with glucose (ANOVA). *—statistically significant difference compared with other groups at basic conditions, for *p* ≤ 0.05; ▲—statistically significant difference compared with other groups in the presence of glucose, for *p* ≤ 0.05.

**Figure 2 biomolecules-11-01885-f002:**
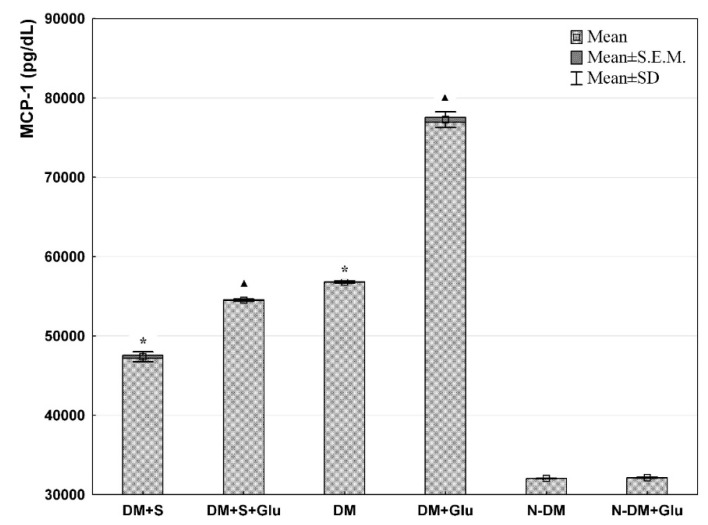
MCP-1 level in the supernatant from a culture of human PBMCs isolated from statin-treated or statin-free type 2 diabetic patients, or from non-diabetic controls in the medium only, or preincubated with glucose (ANOVA). *—statistically significant difference compared with other groups at basic conditions, for *p* ≤ 0.05; ▲—statistically significant difference compared with other groups in the presence of glucose, for *p* ≤ 0.05.

**Figure 3 biomolecules-11-01885-f003:**
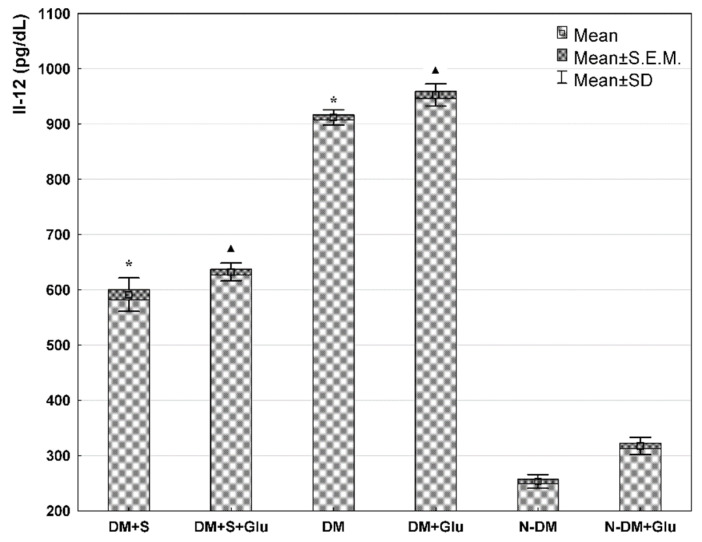
IL-12 level in the supernatant from a culture of human PBMCs isolated from statin-treated or statin-free type 2 diabetic patients, or from non-diabetic controls in the medium only, or preincubated with glucose (ANOVA). *—statistically significant difference compared with other groups at basic conditions, for *p* ≤ 0.05; ▲—statistically significant difference compared with other groups in the presence of glucose, for *p* ≤ 0.05.

**Figure 4 biomolecules-11-01885-f004:**
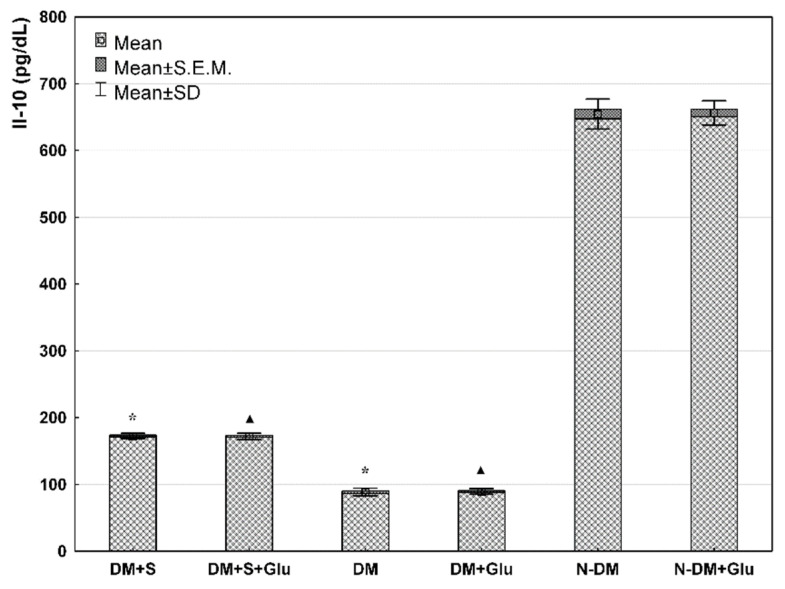
IL-10 level in the supernatant from a culture of human PBMCs isolated from statin-treated or statin-free type 2 diabetic patients, or from non-diabetic controls in the medium only, or preincubated with glucose (ANOVA). *—statistically significant difference compared with other groups at basic conditions, for *p* ≤ 0.05; ▲—statistically significant difference compared with other groups in the presence of glucose, for *p* ≤ 0.05.

**Table 1 biomolecules-11-01885-t001:** Baseline metabolic characteristic of all subjects.

Parameters	DM+S n = 10 (5M/5F)	DM n = 10 (5M/5F)	N-DM n = 10 (5M/5F)
**1,5-AG (mg/L)**	13.5 ± 2.0	14.2 ± 2.7	23.3 ± 2.5
**HbA_1_c (%)**	7.1 ± 0.4	7.1 ± 0.5	5.3 ± 0.4
**Total cholesterol (mg/dL)**	179.7 ± 15.1	240.3 ± 24.2	176.3 ± 19.8
**LDL (mg/dL)**	91.0 ± 16.5	132.1 ± 16.3	101.2 ± 11.5
**HDL (mg/dL)**	54.7 ± 18.6	56.7 ± 16.2	54.9 ± 14.6
**TG (mg/dL)**	144.7 ± 19.1	158.1 ± 16.0	123.4 ± 19.6
Values expressed as Mean ± SD; DM+S—statin-treated patients with type 2 diabetes; DM—statin-free patients with type 2 diabetes; N-DM—non-diabetic control			

## Data Availability

The data presented in this study are available on request from the corresponding author.

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
