# Peer review of "Effects of Low-Dose Atorvastatin on the Peripheral Blood Mononuclear Cell Secretion of Angiogenic Factors in Type 2 Diabetes"

_biomolecules, 2021, doi:10.3390/biom11121885_

Round 1
Reviewer 1 Report
The manuscript is very well written and very clear. The work presented showed promising results, showing that diabetic patients may benefit from the treatment with statins to avoid complex complications such as atherosclerosis-based complications.
Minor points:
Line 74 - It is written "Although previous studies...." but there is only one reference to previous studies. At least 2-3 references should appear.
Line 98 - change the italic formating of mM.
Line 103 - "...-80ºC" should be changed to " -80 ºC (with a space between the number and the unit.
Expression as "in vitro", "in vivo", "et al" must be in italic, therefore authors must make the proper changes in the manuscript
References must be listed using the same font.
Author Response
Point 1: Line 74 - It is written "Although previous studies...." but there is only one reference to previous studies. At least 2-3 references should appear.
Response 1: We added one more reference [6], as below:
Zhou, J.; Cheng, M.; Liao, Y.H.; Hu, Y.; Wu, M.; Wang, Q.; Qin, B.; Wang, H.; Zhu, Y.; Gao, X.M.; Goukassian, D.; Zhao, T.C.; Tang, Y.L.; Kishore, R.; Qin, G. Rosuvastatin enhances angiogenesis via eNOS-dependent mobilization of endothelial progenitor cells. PLoS One. 2013, 8(5), e63126.
Point 2: Line 98 - change the italic formating of mM.
Response 2: The italic formatting has been corrected.
Point 3: Line 103 - "...-80ºC" should be changed to " -80 ºC (with a space between the number and the unit.
Response 3: A space has been added between the number and the unit.
Point 4: Expression as "in vitro", "in vivo", "et al" must be in italic, therefore authors must make the proper changes in the manuscript
Response 4: Formatting has been changed into italic for the expression "in vitro", "in vivo", "et al".
Point 5: References must be listed using the same font.
Response 5: The font for the references has been unified.
Reviewer 2 Report
Dear authors
I have completed the review of the manuscript titled "Effects of Low Dose Atorvastatin on Peripheral Blood Mononuclear Cell Secretion of Angiogenic Factors in Type 2 Diabetes"
This study aims to evaluate the impact of statins on the ability of peripheral blood mononuclear cells (PBMCs) to release pro-angiogenic factors. To this end, the authors collected and cultured PBMCs from diabetic patients treated or not with statins, in the presence of low or high glucose concentrations. PBMCs derived from healthy patients were used as control cells. The authors analyzed the levels of VEGF, MCP-1, IL-10 and IL-12 released into the medium showing that the levels of VEGF, MCP-1 and IL-12 were lower in patients treated with statins while IL-10 (a antiangiogenic agent factor) levels were higher in treated patients than in statin-free patients. Based on these results, the authors suggested that low doses of statins exhibit antiangiogenic activity.
Overall, I find this study interesting. The study is well organized and written, and the results have been clearly shown. However, although the reported data are interesting, I believe that this work is still in a preliminary phase and that the authors need to implement the experimental part to support their hypothesis with further experimental evidence. In essence, the authors show interesting results but their conclusions on the possible mechanisms that determine these changes remain highly speculative, and based only on hypotheses or correlations with other authors' works. For this reason, I believe this manuscript is not yet suitable for publication on Biomolecole.
Major points.
Why did the authors not determine the plasma levels of VEGF, MCP-1, IL-10 and IL-12 in the patients' plasma?
Is it possible that proangiogenic activity was determined only by PBMCs residing in the atherosclerotic plaque?
Can the authors rule out the role of proangiogenic factors also released by different cells?
What are the levels of these cytokines in the blood of patients treated with statins?
Is there a correlation between the in vivo cytokine levels and the in vitro data shown by the authors?
Literature data (lane 217-218) suggest that pro-angiogenic factor concentrations increase in diabetic patients. What is the effect of statins in the patients enrolled in this study?
Lanes 235-240 The authors suggested that statins act as antioxidant agents. The results that support this hypothesis are lacking. The authors should demonstrate that ROS levels decrease in cells derived from statin-treated patients. Why do ROS decrease? Maybe because it increases the expression of scavenger enzymes?
The role of high glucose as a key factor promoting MCP-1 expression, as well as the effects of statins on IL-12 expression and secretion, need to be demonstrated from a molecular point of view.
Author Response
Point 1: Why did the authors not determine the plasma levels of VEGF, MCP-1, IL-10 and IL-12 in the patients' plasma?
Response 1: Authors measured plasma levels of these cytokines in patient’s sera earlier and it was published in two papers [1, 2] cited below. These papers are cited in Discussion chapter cause results obtained in vivo were starting points to the current study.
- Dworacka et. al. Pharmacology.2014;93(1-2):32-8. doi: 10.1159/000357476. Statins in low doses reduce VEGF and bFGF serum levels in patients with type 2 diabetes mellitus.
- Dworacka et. al. Eur J Pharmacol . 2014;740:474-9. doi: 10.1016/j.ejphar.2014.06.041. Circulating monocyte chemotactic protein 1 (MCP-1), vascular cell adhesion molecule 1 (VCAM-1) and angiogenin in type 2 diabetic patients treated with statins in low doses.
Point 2: Is it possible that proangiogenic activity was determined only by PBMCs residing in the atherosclerotic plaque?
Response 2: It is not possible, of course, that proangiogenic activity is determined exclusively by PBMCs residing in the atherosclerotic plaque, but these cells play very essential role in the formation and ultimate structure of atherosclerotic plaque [3], moreover but we did not study atherosclerotic plaque but isolated PBMCs. As it is stated in the Discussion chapter “the role of PBMCs in the angiogenic process seems to be important not only because of their ability to secrete angiogenic factors, but also due to their specific site of action – localization in vascular plaques. This, in turn, may suggest that these cells may significantly modulate the formation of new blood vessels within the atherosclerotic plaque located in the peripheral vessels, thus, affecting the risk of vascular complications.”
- Libby P. Inflammation in atherosclerosis. Arterioscler. Thromb. Vasc. Biol., 32 (2012), pp. 2045-2051.
Point 3: Can the authors rule out the role of proangiogenic factors also released by different cells?
Response 3: Yes, we can rule out the role of angiogenic factors released by cells others than PBMCs, because we used PBMCs isolated from the blood, cultured and only culture supernatants were evaluated for VEGF, MCP-1, Il-10, and Il-12 – see methods chapter.
Point 4 and 5: What are the levels of these cytokines in the blood of patients treated with statins? Is there a correlation between the in vivo cytokine levels and the in vitro data shown by the authors?
Response 4 and 5: As it was explained earlier (see the text above) authors published papers [1, 2] concerning changes of these cytokines concentrations in the blood of patients treated with statins. Directions of changes in these angiogenic factors concentrations in sera of humans treated with stations were compatible with differences observed in supernatants from PBMCs cultures [see papers 1, 2].
Point 6: Literature data (lane 217-218) suggest that pro-angiogenic factor concentrations increase in diabetic patients. What is the effect of statins in the patients enrolled in this study?
Response 6: The effect of statins on serum levels of proangiogenic factors of patients with type 2 diabetes enrolled into the study was described just in two papers mentioned above [1, 2]. Biological material for previously published studies [1, 2] and for the current study was collected at the same time from the same group of patients.
Point 7: Lanes 235-240 The authors suggested that statins act as antioxidant agents. The results that support this hypothesis are lacking. The authors should demonstrate that ROS levels decrease in cells derived from statin-treated patients. Why do ROS decrease? Maybe because it increases the expression of scavenger enzymes?
Response 7: The aim and the conclusion of our study did not concern the potential mechanism throughout which statins decrease the release of pro-angiogenic factors by PBMCs. Since the antioxidant activity of statins was well documented in numerous other publications [e.g. 4,5 – see below] we have grounds, without additional experiment, to discuss in the Discussion chapter the hypothesis that statins develop the anti-angiogenic effect observed by us by the anti-oxidant effects. It is clear, of course, that statins develop the anti-oxidant activity not only by the mechanism related to ROS levels decrease but we discussed only the association between factors affecting angiogenic factors secretion and oxidative stress and found that exactly free radicals were proven as stimulatory factors [6]. Moreover, one of the aims of our study was to evaluate the influence of hyperglycaemia on angiogenic factors released by PBMCs from diabetic and non-diabetic patients in the presence of statins. Since hyperglycaemia is well known factor inducing oxidative stress, especially free radicals formation, and we found that in statin-treated patients the concentrations of pro-angiogenic factors is lower than in statin-free group, we hypothesized (not concluded) in the Discussion chapter that statins are able to limit this deterioration due to hyperglycaemia.
- Wang et al. Pleiotropic effects of statin therapy: molecular mechanisms and clinical results. Trends Mol Med. 2008, 14(1), 37-44.
- Lynn LS et al.Antioxidant effects of statins.Drugs Today (Barc) 2004 Dec;40(12):975-90. doi: 10.1358/dot.2004.40.12.872573. DOI: 10.1358/dot.2004.40.12.872573
- Khatri, J.J.; Johnson, C.; Magid, R.; Lessner, S.M.; Laude, K.M.; Dikalov, S.I.; Harrison, D.G.; Sung, H.J.; Rong, Y.; Galis, Z.S. Vascular oxidant stress enhances progression and angiogenesis of experimental atheroma. Circulation. 2004, 109(4), 520-525.
Point 8: The role of high glucose as a key factor promoting MCP-1 expression, as well as the effects of statins on IL-12 expression and secretion, need to be demonstrated from a molecular point of view.
Response 8: We are not quite sure what is the issue of reviewer’s comment above. Should we also perform the experiment which would explain the role of high glucose as a key factor promoting MCP-1 and Il-12 expression from a molecular point of view? Our study was a type of an observation experiment which aim was to elucidate if statins influence the secretion of angiogenesis mediators just by the peripheral blood mononuclear cells derived from patients with type 2 diabetes, particularly in the presence of acute hyperglycemia. It is very likely that the study of molecular mechanism will be the next step of our study, however the molecular basis of the phenomenon observed by us it was in a part studied and described by other authors and we discussed it in Discussion chapter as below:
“…Glucose, and in particular short-lasting hyperglycemia episodes, is potentially involved in the increased MCP-1 synthesis by PBMCs by means of NF-κB activation [45], as well as an increased generation of free radicals [46]. Statin-related effect, inhibition of HMG-CoA reductase, which is involved in NF-κB signaling pathway [47], may be responsible for the suppression of MCP-1 secretion induced by these drugs. This hypothesis was supported by Diomede et al. [48] in an in vivo experiment, which demonstrated that the inhibition of mevalonate synthesis may influence the MCP-1 production by hepatocytes…”
“…These results are consistent with the previous study indicating glucose-dependent activation of protein kinase C, p38 MAPK (p38), c-Jun terminal kinase, and inhibitory-κB kinase, resulting in the Il-12 secretion by macrophages [55]…”
Round 2
Reviewer 2 Report
No further comments